# Household CO_2_ Emissions: Current Status and Future Perspectives

**DOI:** 10.3390/ijerph17197077

**Published:** 2020-09-27

**Authors:** Lina Liu, Jiansheng Qu, Tek Narayan Maraseni, Yibo Niu, Jingjing Zeng, Lihua Zhang, Li Xu

**Affiliations:** 1Northwest Institute of Eco–Environment and Resources, Chinese Academy of Sciences, Lanzhou 730000, China; liuln@llas.ac.cn (L.L.); Tek.Maraseni@usq.edu.au (T.N.M.); niuyb@llas.ac.cn (Y.N.); zengjj@llas.ac.cn (J.Z.); 2College of Earth and Environmental Sciences, Lanzhou University, Lanzhou 730000, China; lxu16@lzu.edu.cn; 3Centre for Sustainable Agricultural Systems, University of Southern Queensland, Toowoomba 4350, QLD, Australia; 4College of Information, Shanxi University of Finance and Economics, Taiyuan 030006, China; 20151025@sxufe.edu.cn

**Keywords:** bibliometric, review, perspective, household CO_2_ emissions

## Abstract

The household sector, which plays a critical role in emission reduction, is a main source of greenhouse gas (GHG) emissions. Although numerous academic journals have published papers on household CO_2_ emissions (HCEs), great challenges remain in research on assessments, determinants, and further research prospects. This work reviews and projects HCEs using a bibliometric analysis and a systematic review based on the data from the Web of Science (WOS) platform from 1991 to 2020. Over the last 30 years, there has been a rapid and active trend of research on HCEs. We find that (1) the scale of the bibliometric analysis shows that research on HCEs is interdisciplinary and must consider overall cognition of the environment, the economy, society, and technology. It also needs to strengthen cooperation between different countries/territories to emphasize the quality and influence of papers on HCEs. (2) A review of previous literature shows that research on HCEs mainly focuses on the research object, mainstream assessments, and influencing factors. The following six main aspects impact HCEs: demographic, income, social, technological, policy, and natural factors. (3) The research discussion suggests that more micro-level research needs to be conducted, such as research on the city level and the individual level, which is important for sustainable development and low consumption. A comparative analysis of the differences in HCEs is a future research direction. Additionally, localized carbon emission reduction measures need to be implemented.

## 1. Introduction

Global warming caused by anthropogenic emissions since the Industrial Revolution has attracted widespread attention. Despite the increasing public outcry over the last decade, global greenhouse gas (GHG) emissions have risen by 1.5% per year [1]. The main contributors to this increase are China (28%), the USA (15%), the EU (9%), and India (7%), with emission growth rates of +2.3%, +2.8%, −2.1%, and +0.8%, respectively [2]. Human activities have caused approximately 1.0 °C (0.8~1.2 °C) of global warming as compared with the pre-industrial level; this level will reach 1.5 °C between 2030 and 2052 if this rate of increase continues [3].

The Paris Agreement aims to maintain a global average temperature rise below 2 °C and to further limit it to 1.5 °C above pre-industrial levels. However, in 2018, the world reached a record high of 55.3 Gt CO_2_e [1]. Even if all the nationally determined contribution ((NDC) the core of the Paris Agreement) pledges of all countries are realized by 2030, GHG emissions will be 38% higher than the amount needed to limit warming to 1.5 °C [1], and the Earth will be approximately 3.2 °C warmer in 2100 as compared with pre-industrial levels [4]. To realize the 2 to 1.5 °C climate targets, global NDC ambitions need to increase three-fold and five-fold, respectively. Given the political turmoil of climate negotiations, such increases are unlikely [5].

Because of the consumption of a range of goods and services, the household sector is a major contributor to GHG emissions, accounting for more than 60% of global GHG emissions [6,7,8]. From the perspective of different countries/regions, the proportion of household CO_2_ emissions (HCEs) to national emissions is >80% in the USA [9], 30–40% in China [10,11], and 44% in Canada [12]. In 2018, such emissions accounted for approximately 40% of total emissions in Japan [13,14], but direct HCEs accounted for only 4.84% (there were 1078.03 million tons CO_2_ emissions in Japan in 2018, and 52.15 million tons CO_2_ emissions were from the residential sector) [15]. Therefore, realizing the climate target without changing the consumption behavior of household sectors is unrealistic [16,17]. Reducing HCEs is possible and has enormous health and environmental benefits for household residents, but it is necessary to phase out coal and upscale global annual renewable energy investments over the next 30 years (2020/2050) [1].

There is a plethora of research on HCEs, but these studies are not comprehensive or consistent. Many of them considered HCEs only with regard to the consumption of direct goods and services, whereas others considered indirect goods and services [18,19]. Moreover, there are inconsistencies in categorizing direct and indirect emissions. For example, some studies suggested that heating and electricity consumption-related emissions were direct emissions involved in combustion [20], whereas others argued that these emissions were indirect emissions from the indirect consumption [21,22,23,24]. Therefore, a globally standardized rule is necessary. Similarly, some scholars considered only CO_2_ emissions, while others estimated the entire carbon footprint [25]. Therefore, the definitions of CO_2_ emissions in the household sector are not comparable. This paper takes all the above issues as the research object and designates them as HCEs.

Over the last two decades, research on HCEs has grown exponentially, suggesting the growing awareness and recognition of HCEs. However, systematic review papers on HCEs are relatively scarce. In this review, we aim to identify and synthesize papers on HCEs published over a 30-year period (1991–2020). We assess (1) the trend and magnitude of research on HCEs; (2) the journal outlets of the papers; (3) the contributing countries and territories of the papers; (4) the research theme, coverage, and methods used for assessing HCEs; (5) the influencing factors of HCEs; and (6) discussions of the research prospects for HCEs in the papers. On the basis of a comprehensive analysis, we identify many research gaps for further research and areas of global standardization for consistency and comparability among studies. The remainder of this work is organized as follows: In Section 2, we present the bibliometric data and methodology; in Section 3, we provide the research findings; and in Section 4, we offer the discussion and conclusions.

## 2. Bibliometric Data and Methodology

The Web of Science (WOS) platform is an authoritative citation database which contains thousands of international academic journals that are multidisciplinary, comprehensive, and that have a high impact [26]. The data on HCE studies were collected from the Science Citation Index Expanded (SCIE) and Social Science Citation Index (SSCI) databases in the WOS platform. Retrieval was based on the topical subjects Household* OR Resident* AND “carbon emission*” OR “carbon dioxide emission*” OR “CO_2_ emission*” OR “carbon footprint*” OR “carbon dioxide footprint*” OR “CO_2_ footprint*” OR “carbon emit*” OR “carbon dioxide emit*” OR “CO_2_ emit*” OR “Energy emission*” OR “Energy footprint*” was conducted in July 2020. We collected a total of 2303 documents, from 1991 to 2020, based on the retrieval type, including the title, author, keywords, abstract, and citation information.

To more accurately perform the bibliometric data analysis, we cleaned information such as the countries, institutions, and keywords in the abovementioned document data. On the basis of the cleaned literature data, this paper presents a bibliometric analysis of research on HCEs. We determine the status of research on HCEs and provide an overview of development by analyzing published trends, subject distributions, and journal evaluations. We discuss the situation of cooperation of countries and institutions in the HCE field by performing a social network of such cooperation. We capture the high-frequency keywords in HCE research to reflect the status quo and the main research directions. Derwent Data Analyzer (DDA, Clarivate Analytics, Boston, MA, USA) and VOSviewer version 1.6.13 (Centre for Science and Technology Studies, Leiden University, Leiden, The Netherlands) were used in this study to extract general statistical information and to visualize the network.

The number of published papers on HCEs shows an upward trend, indicating that awareness of HCEs has been growing [27,28]. However, research on HCEs has involved a wide range of topics, and systematic quantitative analysis was lacking. There is an urgent need to identify related research in the HCE field. Regarding the academic papers, quantitative analysis of the progress of research on HCEs using a systematic bibliometric method can help researchers to identify research prospects and trends. A comprehensive review of HCEs must consider research methods, the related influencing factors, and future policy recommendations, as well as future research prospects. This work conducts a bibliometric analysis and literature review on HCEs to present an in-depth study of the impact mechanisms and future research trends.

## 3. Research Findings

### 3.1. Research Status

#### 3.1.1. Overall Publishing Trend

From 1991 to 2020, a total of 2303 articles conformed to the criteria for research on HCEs, as shown in Figure 1. The number of articles increased from 1 in 1991 to 330 and 215 in 2019 and 2020, respectively. The papers published in 2020 were relatively new (the paper retrieval time was up to July 2020), causing a certain lag phenomenon (Figure 1). However, this aspect does not affect the overall analysis because a linear growth trend (with R^2^ = 0.9104) was observed based on the number of annual articles from 1991 to 2020. There are three general phases of analysis of HCEs. In the first phase, from 1991 to 2000, the number of published articles on HCEs accounted for 1.91% of the total published articles, with an average number of only 4.40. By 2000, the cumulative number of published articles was only 44. This stage could be considered the initial stage of research on HCEs. In the second phase, from 2001 to 2010, research on HCEs accounted for 10.90% of the total published papers, with an average number of 25.10. By 2010, the cumulative number of published papers was 251, an increase of 4.70 times as compared with the 2000 level. Research on HCEs has maintained a slow but steady growth rate, indicating that HCEs have gradually received attention. In the third phase, from 2011 to 2020, research on HCEs accounted for 87.19% of the total number of papers published, with an average number of 200.80. In 2019, the number of published articles was 330. The cumulative number of published articles increased until 2008, an increase of 7.00 times as compared with the 2010 level. During the past 30 years, the overall publishing trend of research on HCEs has been growing, revealing that studies on HCEs have received widespread attention.

#### 3.1.2. Journal Outlets of the Paper

Studies on HCEs were published in 430 different journals. The top 15 most productive journals for research on HCEs accounted for 53.44% of the total publications, with each journal publishing no fewer than 27 papers, as shown in Table 1. Among them, *Energy Policy* ranked as the top HCE-related journal, with the highest percentage (8.99%), followed by *Journal of Cleaner Production* (8.00%), *Energy and Buildings* (6.49%), *Applied Energy* (5.37%), *Sustainability* (4.82%), and other journals. This result implies that these are the key journals in HCE research. Interestingly, regarding the distribution of the top 15 productive journals, the journal distribution is relatively concentrated, mainly in the UK, the Netherlands, Switzerland, the USA, and Germany. This is consistent with the fact that these countries are at the forefront of this research field.

#### 3.1.3. Contributions of Countries/Territories

During the past 30 years, papers on HCEs have involved 91 different countries/territories. The top 15 productive countries/territories with the most publications on HCEs accounted for 76.01% of the total, with each publishing more than 53 papers, as shown in Figure 2. The countries include China (675, 21.05%), the USA (412, 12.87%), the UK (288, 9.00%), Japan (149, 4.65%), Australia (137, 4.28%), Spain (116, 3.62%), Canada (94, 2.94%), Germany (93, 2.91%), Italy (86, 2.69%), the Netherlands (85, 2.66%), Sweden (77, 2.41%), South Korea (62, 1.94%), Austria (53, 1.66%), Finland (53, 1.66%), and Switzerland (53, 1.66%). Regarding the publications over the past thirty years, the number of articles published by the top 15 countries has been continuously increasing. The USA, the UK, and Austria published earlier and have maintained a lead in HCE research. The number of publications by the USA (as a developed country) and China (as a developing country) is significantly higher than that by other countries. The main reason is that rapid economic development in the USA and China has led to improvements in quality of life, as well as increased energy usage and the related HCEs. China published HCE papers later, but it has rapidly developed and is now the country with the most publications. After 2012, the number of published articles in China surpassed that from the USA, with the significant advantage of being a latecomer. Additionally, HCEs accounted for >80% of the total emissions in the USA [9] and 30–40% in China [10,11], and CO_2_ emissions from the household sector have aroused widespread concern abroad [27]. Reducing HCEs will make a significant contribution to global climate action [28]. The USA, China, the UK, and Australia are the main regions where HCE research is active.

The trend of HCE research publications is related to climate change events that are important for the international community. For example, the United Nations Framework Convention on Climate Change (UNFCCC), adopted in June 1992, was the first international convention to control GHG emissions and combat climate change. Among the top 15 productive countries, the USA and the UK published one document related to HCEs in 1993. The Kyoto Protocol, adopted in December 1997 as a supplement to the UNFCCC, makes GHG reduction a national legal obligation. Developed countries such as the USA, the UK, Australia, Canada, and Austria began to pay attention to climate change and conducted research in the field of HCEs, leading to a slight increase in the number of publications. At this time, the world had not yet formed a unified and solid emission reduction action and climate negotiations agreement, and there was less concern about the national interests of various countries. Therefore, international attention to HCE research was generally not high.

The Delhi Declaration in 2002, the Bali Roadmap in 2007, and the Copenhagen Accord in 2009 all had epoch-making significance. Since then, both developed and developing countries have made commitments to reduce emissions in the future. The Paris Climate Agreement was adopted in December 2015, with an increase of 1.5 °C as the target to address climate change. At various time points, the number of publications on HCEs by various countries have increased at different levels, indicating that HCE research in various countries is affected by climate policies. Especially after 2010, both the developed and developing countries among the top 15 productive countries show a clear upward trend in HCE publications, indicating that global emission reduction and climate change mitigation have received widespread attention to ensure global action. As the world’s largest developing country and a major party in climate negotiations, China bears a major responsibility and faces great pressure to reduce emissions. At the Paris Climate Change Conference in December 2015, China proposed that it would attempt to achieve peak CO_2_ emissions by approximately 2030. This proposal shows that China is increasingly interested in HCE research, which is why, since 2012, it has produced more publications on HCEs than any other county. Understanding and evaluating low-carbon development from the HCE perspective are important ways for China to cope with the increasing pressure with regard to emissions reduction and climate negotiations.

The national cooperation network map of HCE research clearly reflects the central scale and concentration in different countries and effectively identifies the intensity of cooperation of various countries. This work draws national cooperation network maps for HCE research using VOSviewer software (Centre for Science and Technology Studies, Leiden University, Leiden, The Netherlands) (Figure 3a) and an online drawing platform named tubiaoxiu [29] (Figure 3b). From the perspective of country centrality, the characteristics of national cooperation in HCE research show a “three-core, multicenter” characteristic distribution centered on China, the USA, and the UK. This result further demonstrates the leadership of developed countries, represented by the USA and the UK, and developing countries, represented by China, in reducing GHG emissions and addressing climate change. Germany, Australia, Japan, Canada, and Sweden are other central countries in this field; they have also cooperated closely to make important contributions to HCE research. China, with the most publications on HCEs, holds a certain advantage in this research field. However, due to the relatively insufficient output of high-quality papers, China may not be sufficiently recognized by the academic community. It is necessary to consider cooperation among developing and developed countries in academic research, for example, on carbon labeling schemes [30], and market mechanisms [31]. Thus, there is an urgent need to strengthen cooperation among China and other countries, especially USA, the UK, and Australia, to enhance China’s influence on HCE research.

#### 3.1.4. Research Category

As identified by the WOS platform, studies on HCEs involve 105 different categories. The top 15 productive categories of HCE research account for 87.35% of the total investigated publications, as shown in Figure 4. Among them, the environmental science category ranks as the top HCE-related category, with the highest percentage (19.66%). Other important research fields include energy and fuels (19.02%), environmental studies (12.01%), green and sustainable science and technology (10.12%), and economics (8.46%). HCEs are an interdisciplinary research topic involving energy, the environment, economics, green development, climate change, health, transportation, and other disciplines in the natural sciences. Therefore, HCE research can provide a comprehensive understanding of the overall knowledge of the environment, the economy, society, and technology.

#### 3.1.5. Hotspots

Keywords can represent the research subjects of academic articles, and bibliometric analysis of keywords can provide an in-depth understanding of the research hotspots and directions in HCE research. There are 5723 keywords in the HCE literature, of which 4537, 580, and 222 keywords have appeared once, twice, and three times, respectively. Keywords with ≤three occurrences account for 93.29% of the total keywords. There are 384 keywords that appear more than four times, accounting for only 6.71% of the total. These keywords are considered to be the main hotspots and research directions of HCE research.

To clarify the development and forefront of HCE research, this study uses VOSviewer software (Centre for Science and Technology Studies, Leiden University, Leiden, The Netherlands). to draw a map of the keyword co-occurrence network for keywords with an occurrence frequency of more than 40 times, which removes the subject keywords (such as carbon emissions, CO_2_ emissions, and carbon footprints) (Figure 5a). The topical subjects, such as consumption, energy efficiency, GHG emissions, and life cycle assessment (LCA), are analyzed. The co-occurrences between these keywords are very close. First, regarding the topic of household consumption, the main focuses are energy types, energy conservation, energy policies, and input-output analysis in different industries and sectors (such as construction and transportation). Second, the subject of energy efficiency mainly focuses on the construction sector, the household sector, energy conservation, carbon emission reduction, LCA, and sustainable development. This topic mainly explores the carbon emissions status and low-carbon policies as key issues. Third, GHG emissions are integrated into the theme of climate change, focusing on the global warming caused by energy consumption as well as the status and development trends of renewable energy, carbon sequestration, and carbon trading. Fourth, China is highlighted as a research area; this research mainly focuses on models such as stochastic impacts by regression on population, affluence and technology (STIRPAT) and structural decomposition analysis (SDA) to analyze the relationship between urbanization and energy consumption. There is a strong correlation between these four themes, which shows that different areas and different directions of HCEs are mutually inherited and connected.

Burst detection is used to determine whether there has been any change in the research hotspots, and it can help to gain insights into future research topics [30,32]. We identify the keywords ranked by burst detection (the red rectangle means the strongest bursts), as shown in Figure 5b. In this study, there are 25 keywords with apparent bursts. This result implies that HCE research has been distinguished by the following three stages: In the first stage (2001–2010), as the infancy of HCE research, the burst keywords mainly included energy use, CO_2_ emissions, industrial ecology, climate change, cost, household consumption, energy requirement, sustainable consumption, electricity, India, the USA, and energy demand. In particular, HCE research was gradually extended to direct energy usage and indirect household consumption to evaluate emissions. In the second stage (2012–2014), the main keywords were industry, trade, input-output analysis, renewable energy, simulation, technology, the UK, construction, and environmental impact. In this stage, studies focused on HCE methods and the possible impact of HCEs on the environment as well as the related influencing factors, such as trade, technology, and renewable energy usage. The third stage, from 2016 to 2018, involved research on HCEs at the city level and focused on sustainability and urbanization. HCE research shifted from the whole country (such as in India, the USA, the UK) scale to the micro scale, such as the city level.

According to the above bibliometric analysis, the research topics in the HCE field mainly include the following three aspects. First, basic research on HCEs focuses on descriptions and measurements, including the research contents, methods, application areas, and impacts on climate change. Second, mechanism analysis focuses on the simulation of resource coupling, including carbon emissions, carbon footprint, and GHG emissions reduction, as well as global climate change mitigation, adaptation mechanisms, and technological development. Third, application research focuses on scenario analysis on decision making and HCE implementation methods, including energy efficiency, energy policy, environmental impact assessment, and other methods and theories in different countries (such as the USA, China, and the UK) in different fields (such as agriculture, industry, transportation, electricity, biomass, and waste disposal).

### 3.2. Research Review

#### 3.2.1. Research Subjects

In the related research literature, there are different expressions for HCEs. For example, CO_2_ emissions are emitted from household/residential direct energy usage and indirect household/residential consumption [21,22,23,24]. Carbon/CO_2_ footprints are emitted from the direct energy combustion of households/person and the indirect consumption of goods and services [25]. Research considers CO_2_ emissions or footprints that only come from direct household/residential energy usage, from indirect household consumption, or from one or more sources of energy, products, or consumption [13,14,18,19,20]. Although the above expressions are different, they all refer to CO_2_ emissions generated by direct energy usage or indirect consumption from the household/residential sector [33]. Therefore, this paper includes all the above expressions of HCEs in the research coverage.

HCEs are classified based on different energy types, different life demands, and different consumption behaviors (Figure 6). From the perspective of different energy types, HCEs can be divided into primary energy HCEs (referring to CO_2_ emissions generated by direct primary energy usage, such as coal, gas, and oil), secondary energy HCEs (referring to CO_2_ emissions generated by the indirect combustion of energy usage, such as household electricity and heating), and household consumption HCEs (referring to CO_2_ emissions generated by indirect household energy consumption, such as food, clothing, housing, household equipment, medical care, transportation, communications, culture, education, entertainment, and other services) [21,22,23,24,34]. From the perspective of different life demands, HCEs can be divided into basic demand HCEs (referring to CO_2_ emissions generated by energy consumption for the activities necessary to meet basic living demands, mainly including direct energy usage from coal and gas and indirect energy consumption from electricity, heating, food, clothing, and housing) and development demand HCEs (referring to CO_2_ emissions generated by energy consumption for developmental activities to meet development living demands, mainly including direct energy usage from gasoline and diesel and indirect energy consumption for cultural, educational entertainment, transportation and communication, and other services). From the perspective of different consumption behaviors, HCEs can be divided into the following five categories: clothing HCEs, food HCEs, housing HCEs, transportation HCEs, and service HCEs [21,22,23,24].

Reviewing the HCE literature, this study analyzed the basic progress in research on HCEs. The main HCE research hotspots have shifted from CO_2_ emissions from direct household energy usage and indirect household consumption to one category, such as gas [35], electricity [36], heating [37], and residential housing [38,39]. In general, the research subjects have shifted from the whole to the individual. Future research on HCEs needs to focus on studies from a behavioral perspective.

#### 3.2.2. Research Methods

In the development of HCE research, the greatest contribution has been made by the development of complete computing methods, including the consumer lifestyle approach (CLA) [9], the IPCC’s Reference Approach (IPCC-RA) [40,41], the input-output method (IOM) [18,42], and LCA [36]. For example, Zhang et al. (2015) reviewed the quantification methodologies of HCEs and showed that the above four methods were the mainstream methods [27]. Geng et al. (2017) summarized four methods, including behavioral analysis, LCA, cost-benefit analysis, and IOM, by filtering subject words [28]. This paper briefly described the mainstream assessments of HCEs and their scope of application, which was suitable for different research objects and scales.

LCA is usually used to calculate the GHG emissions generated in the whole process of product production and service through the entire life cycle. It is mainly used in household buildings and transportation [36]. However, LCA requires more precise data, which makes it difficult to measure HCEs on a large scale. IPCC-RA can be applied to assess direct HCEs from macro statistics or micro survey data of various countries, territories, provinces and cities [12,40,41]. The advantage of IPCC-RA is that the calculation is simple and that the method is easy to understand and operate. However, due to the different energy production levels in different regions, there are significant differences in the carbon emission coefficients of the same energy source. Therefore, future HCE research needs to focus on the differences in emission coefficients of different countries, regions, and cities to obtain more accurate and comparable data. The IOM is mainly applied to evaluate indirect HCEs from macro statistics data or micro survey data of cities in various countries and regions [18,42,43,44]. IOM calculation results are more comprehensive and accurate. However, due to the absence of city or community input-output tables, the indirect HCE coefficients of many cities or communities are based on the entire country or region, causing errors. Future research in this area needs to focus on compiling input-output tables for cities or communities to obtain more detailed micro scale HCE data. CLA is typically used to calculate the indirect HCEs generated by household consumption by multiplying household consumption expenditure by the IOM emission coefficients [9,10,45]. CLA can be applied to assess indirect HCEs from macro statistics or micro survey data and is crucial for understanding the determinants of indirect HCEs.

The accuracy of HCE measurements can be improved by combining the IOM and CLA. It is necessary to combine the above HCE assessments to effectively measure HCEs and provide data support to address low-carbon development at the city level. Hybrid models of HCEs are urgently needed to address local climate change governance.

#### 3.2.3. Influencing Factors of HCEs

In recent research on HCEs, an increasing number of scholars have analyzed the influencing factors, difference comparisons and mitigation policies regarding HCEs to understand the relationship between HCEs and population, resources, the environment, the economy, and technology [9,10,21,22,23,24]. Mechanism analysis in HCE research is crucial for formulating corresponding low-carbon emission reduction strategies within various countries, regions, or cities. The literature review on HCE research showed that research on the influencing mechanism of HCEs mainly included demographic, income, social, technological, policy and natural factors, as summarized in Figure 7, and is described as follows:The demographic factors of HCEs are analyzed from the perspective of the population structure (such as the gender and age structure), population density (such as housing density and land density), and population quantity (such as household size, total population, and urbanization). The assessment of India by Rosenberg et al. showed that women were neither the only nor the main beneficiaries of electricity [46]. Ota et al. noted that the aging of society and the population both decreased and increased Japan’s electricity demand but did not increase or decrease its gas demand, respectively [34]. Chancel showed that baby boomers in France emitted more HCEs than other generations, while there were no generational effect in the USA [47]. Yu et al. noted that as consumption patterns changed, the shift to smaller and aging households produced more household energy usage and carbon emissions [48]. Different researchers have different views on this mechanism of HCEs.The income factors of HCEs include the income level (such as income per household, income per capita, and total income) and the consumption level (such as consumption ability and consumption tendency). Some results show that income and consumption play an important role in increasing HCEs. For example, an increase in per capita income resulted in increased HCEs in China [21,45,49], Ireland [50], France, and the USA [47]. Wiedenhofer et al. and Wu et al. showed that there was inequality in household carbon emissions and household energy usage [25,38]. The C40 Cities indicates that global GHG emissions mainly come from urban consumption [51].The social factors of HCEs include economic development (such as total GDP and GDP per capita), the education level (such as the number of educated people and the number of universities), and lifestyle (such as culture and social awareness). Some arguments emphasize that more energy usage and related HCEs increase as the economy grows [45]. Li et al. reviewed HCEs on a scale of social awareness and lifestyle, which play important roles in HCEs [52]. Hafner et al. found that promoting behaviors such as social norms and habits could reduce HCEs from thermal energy demand [53]. Sköld et al. found that people needed to carry out a moderate change in their lifestyle in regard to mobility, which could help to achieve a substantial reduction of 50% [54]. Meangbua et al. showed that in Thailand, education positively impacted direct household CO_2_ requirements and negatively impacted indirect household CO_2_ requirements [55]. Therefore, social factors have different impacts on HCEs.The technological factors of HCEs include emission intensity (such as HCEs per unit GDP and HCEs per unit consumption) and technology application (such as innovation, investment, and professional skill). Some have argued that higher living standards and technology levels led to higher per capita household energy consumption due to the potential rebound effect in energy efficiency and technology [56]. However, others have disagreed [57]. Asumadu-Sarkodie et al. argued that low carbon technology and cleaner energy transition could help to alleviate environmental pollution [57].The policy factors of HCEs include incentives (such as encouraging subsidy policies and political policies) and punitive policies (such as punitive subsidy policies and political policies). An analysis of the carbon footprint of the welfare state shows that green investments can reduce the carbon footprint with no unnecessary rebound effects [58]. Zhang et al. summarized incentive measures and punitive measures for HCEs and found that different local environmental policies for HCEs in different countries, regions, provinces and cities were crucial for energy conservation and carbon emission reduction [27].The natural factors of HCEs include resources (such as water and land), the environment (such as the air quality index (AQI) and refuse disposal), climate (such as local climate and extreme climate), and energy (such as direct energy usage and indirect energy consumption). Studies on the impact of natural factors on HCEs have mainly focused on the relationships among HCEs and climate change. For example, Nie et al. found that the climate effects caused by abnormal temperatures led to income growth, which led to the use of more household energy consumption and the production of more HCEs [59].

## 4. Discussion and Conclusions

### 4.1. Future Perspectives and Discussion

#### 4.1.1. More Micro-Level Research on HCEs and Further In-Depth Mining

Research on HCEs from a micro-level (e.g., the city, community, or individual level) is important for sustainable development and low-carbon consumption. Research on the evolutionary characteristics of HCEs is mainly focused on trends in temporal variations, the characteristics of the spatial distribution, and differences in the spatiotemporal evolution at the macro scale, such as the global, national, regional and provincial levels. However, HCE research on a micro scale, such as the city [60,61,62], community [63], or individual level [64], is relatively lacking in terms of both statistics and survey data. The city is the key node nationally and obtaining accurate city-level HCE data is the first step in achieving a low-carbon economy, sustainable development, sustainable consumption, and other emission reduction policies [65].

Additionally, research on HCEs at the micro-level scale needs to pay attention to the following three aspects. First, the micro-level calculation model for HCEs needs to be optimized and modified, even though the mainstream methods of analyzing HCEs are very mature. This need mainly stems from the large differences in the technological level and production capacity of different countries and regions, resulting in inconsistent emission factors in the production and consumption of fossil fuels [66]. When calculating HCEs on a micro scale, emission factors need to be considered to make the calculated data more reasonable and comparable [67]. Second, a globally standardized rule is necessary for different expressions of HCEs. As mentioned above, the definitions of HCEs are different, especially based on the micro scale, which makes it difficult to compare research results. Creating such a rule is necessary for different research purposes and data availability, for example, by reviewing the HCE literature of the international community and establishing the definition and boundary. As shown in Figure 6, HCEs are classified based on different energy types, different life demands, and different consumption behaviors. Accordingly, a simple and practical HCE calculator based on different levels for diverse stakeholders is needed. Third, the research perspectives on HCEs need to be refined, and data need to be mined. Against the backdrop of big data, a wide variety of data, such as remote sensing satellite data (Landsat data, data from the Defense Meteorological Satellite Program’s Operational Linescan System (DMSP/OLS), data from the Suomi National Polar-Orbiting Partnership’s Visible-Infrared Imaging Radiometer Suite (NPP/VIIRS), etc.) [68,69], statistical data (statistical yearbooks, statistics, etc.) [10,70], and survey data [20,22] are used to research HCEs. As a result, HCE data are diverse, complicated, and heterogeneous and requires in-depth analysis and data mining.

Micro-level research on HCEs is an important direction for the future accounting and management of carbon emissions. Due to the lack of micro-level data, it is very difficult to reasonably assess HCEs on a micro scale. Therefore, it is important to conduct research on HCEs at the city, county, community, and individual level or as a grid scale based on micro scale data (including statistical data, survey data, remote sensing data, and night light data), which can narrow the research scope to a more detailed scale [71].

#### 4.1.2. Comparative Analysis of the Differences in HCEs Is a Future Direction

The theoretical system and research methods of HCEs are becoming increasingly mature. However, research on HCEs based on comparative analyses of international/regional/city differences is lacking. The above assessment methods and impact mechanism analysis show that scholars perform basic calculations using HCE data from different regions, different types, and different sources at various research scales and analyze the evolutionary characteristics from different research perspectives. For example, some important studies have investigated the evolutionary characteristics of HCEs in China, the USA, the UK [72], Japan, India, Finland, the Netherlands, and Australia [73]. HCEs are determined by different factors in different countries/territories, where different driving forces need to be considered. Comparing the differences will clarify the driving mechanism, such as different commuting habits, policies, or levels of public awareness of low carbon. Additionally, research on HCEs should focus on these differences in different categories, such as developed and developing regions (based on different economic levels).

Comparative research on HCEs regarding different consumption behaviors requires extensive attention. The urban population has increased substantially with the rapid progress of urbanization, causing a continuous increase in direct energy consumption by households for cooking, traveling, heating, and other household activities [70]. Additionally, with improvements in household living conditions and quality of life, indirect energy consumption from various products and services is inseparable from the direct energy supply. HCEs mainly come from CO_2_ emissions that are active in indirect household consumption. Consumption actions in household areas, such as clothing, food, buildings, transportation, and household appliances, can significantly reduce CO_2_ emissions and provide more benefits to residents. Furthermore, the relationship between HCEs and all kinds of “flows”, such as energy types and behaviors, should be considered.

#### 4.1.3. Emission Reduction Measures Need to Be Localized

The keyword co-occurrence analysis shows that HCE mitigation measures include improving energy efficiency, advocating energy conservation, and changing consumer behavior. However, many mechanisms influence HCEs. It is necessary to further explore the impact mechanism of HCEs by combining natural factors and human activities. Furthermore, it is necessary to develop models of the influence mechanisms of HCEs, including temporal, spatial, and spatiotemporal perspectives.

Elaborating the characteristics and determinants of the various HCE reduction measures lays the foundation for quantitative research on policy effects. Human activities have major impacts on HCEs, and there are significant differences in HCEs between countries, regions, provinces, and cities. The carbon emissions gap is the focus of the dispute over low-carbon reduction in various countries and regions [31,74]. An understanding of the regional differences in HCEs can contribute to an in-depth understanding of HCEs. Policymakers need to consider the spatial heterogeneity of each country and propose local emission reduction measures based on the actual situation of each region. Therefore, the most effective measures for reducing emissions need to be localized and implemented.

Scholars have gradually paid increasing attention to HCEs reduction, with a rich research perspective that lays the foundation for mitigation and adaptation to climate change. Due to the complexity of carbon reduction and the response to climate change, no single method simultaneously can address all of the related population, social, economic, and ecological issues. Consequently, it is critical to propose more innovative mitigation policies that integrate different methods.

#### 4.1.4. Limitations

The present study uses bibliometric analysis on the progress of HCE research over the past thirty years. The HCE publication data were collected from the WOS platform, which is relatively popular and comprehensive. This work analyzes the overall publishing trend, the contributions of categories, journals, countries/territories and institutions, and the topical subjects of HCE research to explore the progress and frontiers of such research. However, this work has some limitations that can be improved upon in the future. For example, this work mainly includes literature from the WOS platform in English, resulting in an incomplete analysis. Future work should consider and compare non-English literature from different platforms, such as WOS, Scopus, and others. Additionally, the factors of total citation frequency and citations per paper, which are not relevant to the topic of this paper, are not considered. How can a clearer theoretical framework be established in this field, and what is the general research paradigm for such a framework?

## 5. Conclusions

The direct energy usage and indirect energy consumption of households have attracted extensive attention from the international community. By conducting a bibliometric analysis of the international HCE literature, this study gains a comprehensive macro understanding of the research methods, impact mechanisms, and main research directions of this research field. On the basis of a quantitative bibliometric analysis and a qualitative literature review, the progress and frontiers of HCE research mainly include the following aspects:We find that research on HCEs has shown a rapid and active trend over the last 30 years that is highly consistent with national action on climate change and carbon emission reduction. After the Copenhagen Accord in 2009, the number of HCE papers published in 2010 increased significantly. From the perspective of country contributions, research on HCEs is mainly performed by China, the USA, and the UK. It is necessary to strengthen the emphasis on the quality and influence of papers by strengthening cooperation between China and other countries, especially the USA, the UK, and Australia.According to the keywords of international HCE papers, the main topics are relatively concentrated and focus on the subjects of energy efficiency, climate change, CO_2_ emissions, and energy consumption. Scholars first focused on the direct research field of HCEs, including direct energy usage from coal, gas, and oil, and then focused on the analysis of HCEs with regard to the influencing factors, difference comparisons, and mitigation measures.Three types of HCE research progress, including categories, mainstream assessments, and influencing factors, are analyzed. Research on HCEs from a micro level is an important direction that is crucial for sustainable development and low-carbon consumption. Regarding the influencing mechanisms of HCEs, six aspects are summarized which include demographic, income, social, technological, policy, and natural factors.With regard to the prospects for HCE research, we find that three aspects need to be considered. More micro-level research on HCEs needs to be conducted. On the one hand, the micro-level calculation model for HCEs needs to be optimized, and data need to be mined. On the other hand, a globally standardized rule for the HCE framework is necessary. Additionally, carbon emission reduction measures for HCEs need to be localized. National-, regional-, provincial-, and city-level, low-carbon emission reduction policies must be proposed, such as improving household energy efficiency, reducing the intensity of HCEs, and improving household consumption lifestyles, to provide a scientific basis for local climate change governance. Pioneering research on HCEs holds great significance for systematically understanding low-carbon policy, socioeconomic environmental impacts, and other aspects.

## Figures and Tables

**Figure 1 ijerph-17-07077-f001:**
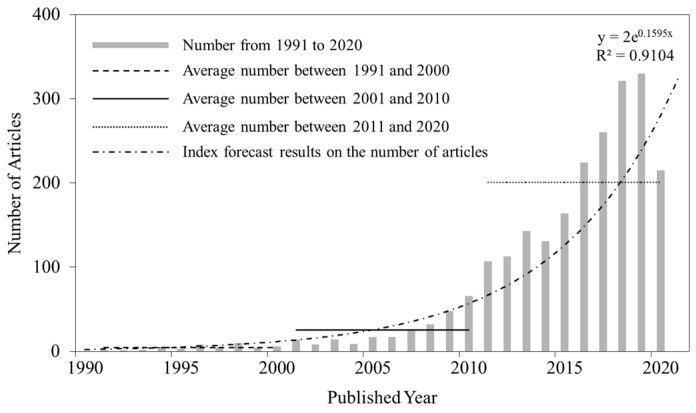
The number of annual research articles on household CO_2_ emissions (HCEs) from 1991 to 2020.

**Figure 2 ijerph-17-07077-f002:**
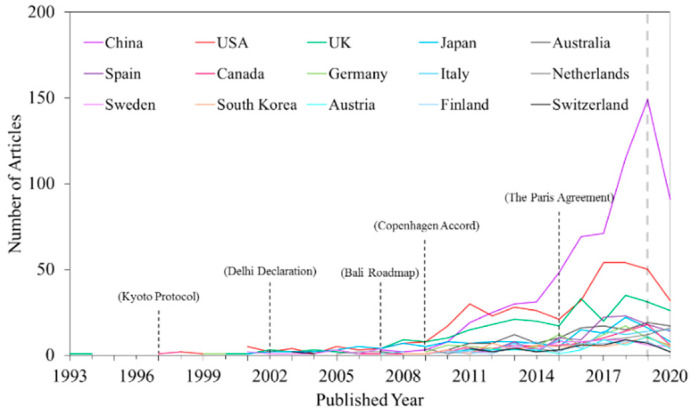
Contributions of the top 15 countries/territories to HCE research.

**Figure 3 ijerph-17-07077-f003:**
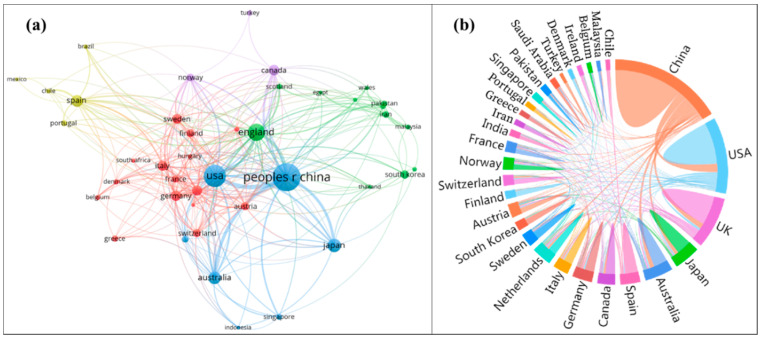
National cooperation network maps for HCEs research from 1991 to 2020. (**a**) Co–authorship among countries; (**b**) cooperation among top 30 countries/territories.

**Figure 4 ijerph-17-07077-f004:**
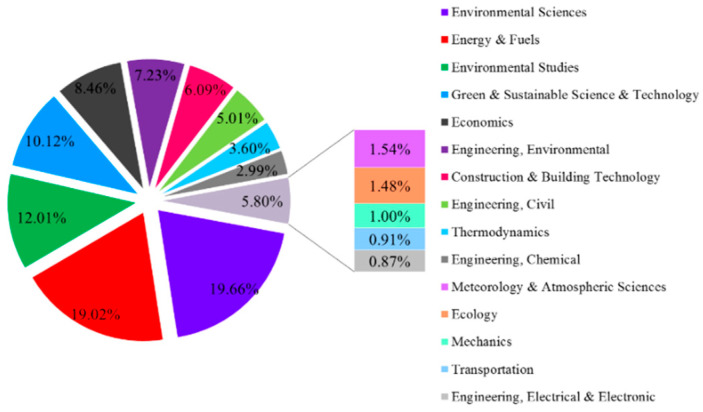
Top 15 productive categories of HCEs.

**Figure 5 ijerph-17-07077-f005:**
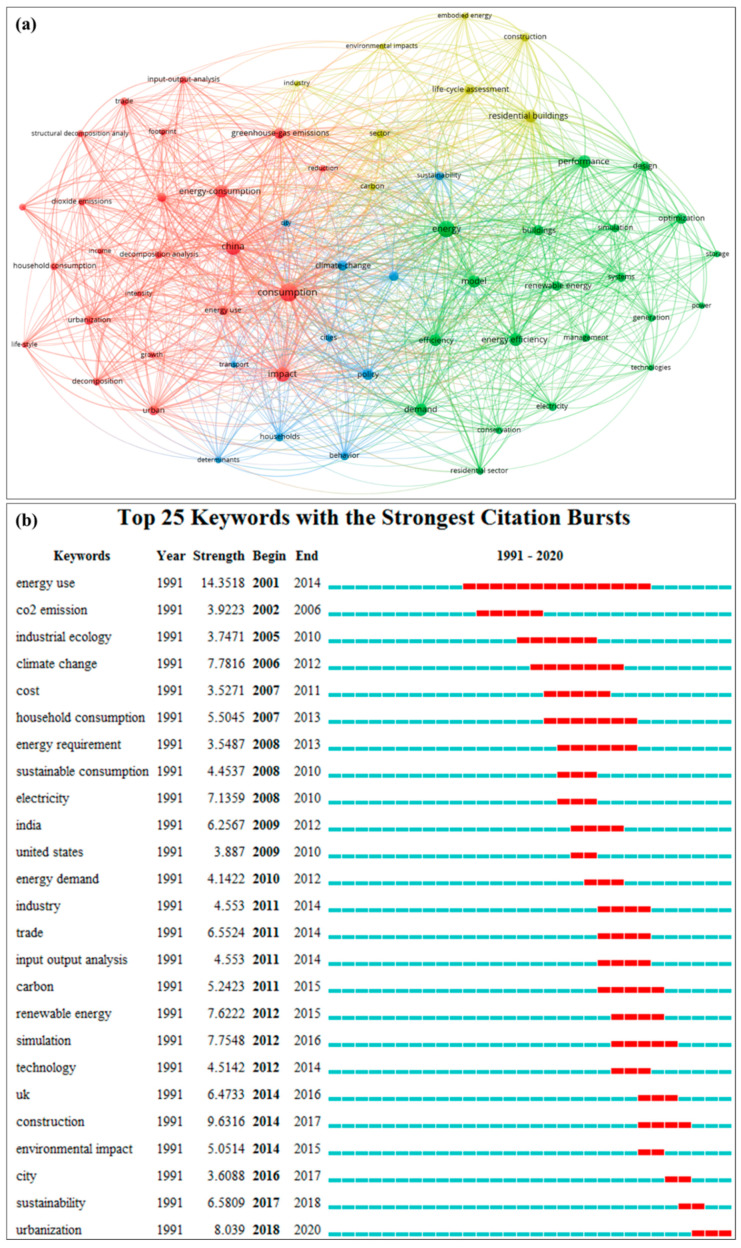
(**a**) Keyword co-occurrence network map without research subjects; (**b**) Keywords ranked by burst detection.

**Figure 6 ijerph-17-07077-f006:**
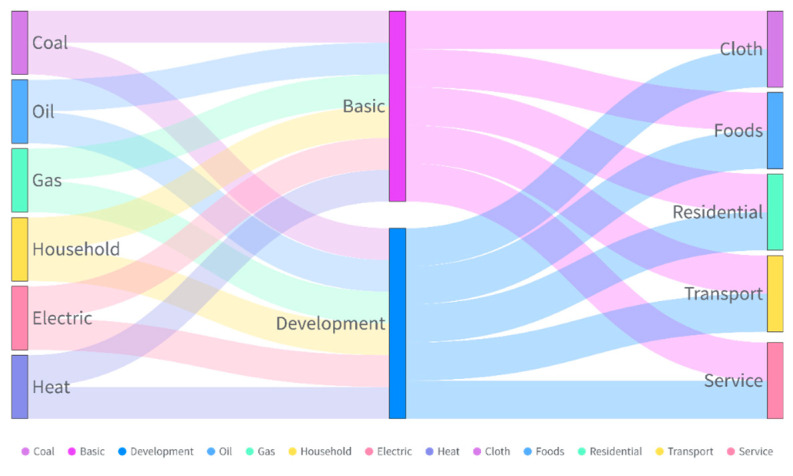
The main research subjects for HCEs.

**Figure 7 ijerph-17-07077-f007:**
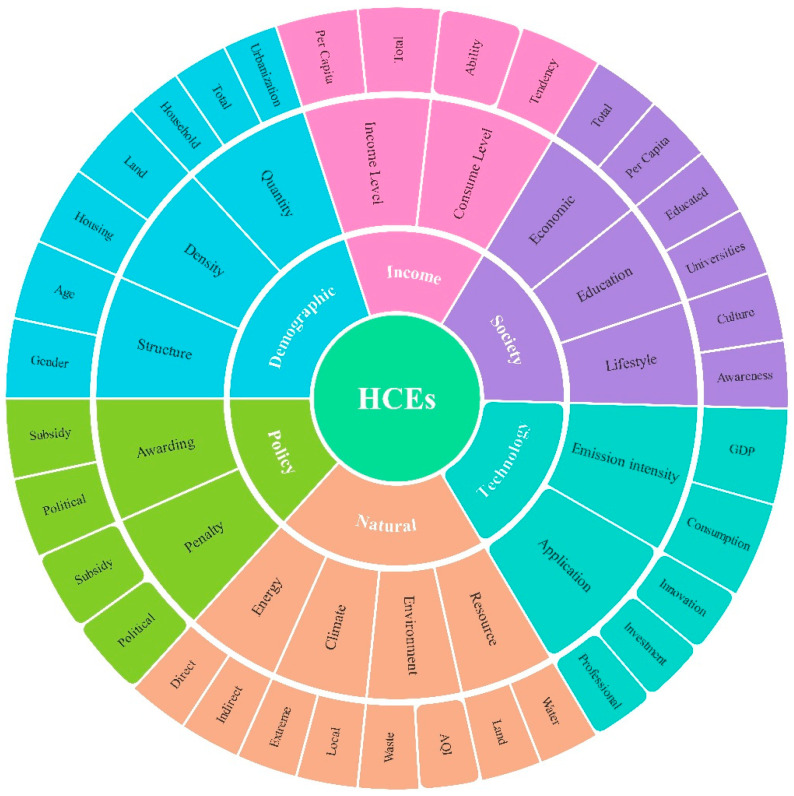
The main influencing factors of HCEs.

**Table 1 ijerph-17-07077-t001:** The top 15 productive journals in HCE research.

Rank	Journals	Number	Ratio (%)	Impact Factor	Country
1	*Energy Policy*	209	8.99	5.042	UK
2	*Journal of Cleaner Production*	186	8.00	7.246	UK
3	*Energy and Buildings*	151	6.49	4.867	Switzerland
4	*Applied Energy*	125	5.37	8.848	UK
5	*Sustainability*	112	4.82	2.576	Switzerland
6	*Energy*	104	4.47	6.082	UK
7	*Energies*	62	2.67	2.702	Switzerland
8	*Energy Economics*	56	2.41	5.203	Netherlands
9	*Ecological Economics*	49	2.11	4.482	Netherlands
10	*Building and Environment*	44	1.89	4.971	UK
11	*Environmental Science & Technology*	31	1.33	7.864	USA
12	*Sustainable Cities and Society*	30	1.29	5.268	Netherlands
13	*Energy Conversion and Management*	29	1.25	8.208	UK
14	*Environmental Science and Pollution Research*	28	1.20	3.056	Germany
15	*Science of The Total Environment*	27	1.16	6.551	Netherlands

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
