# Peer review of "Household CO2 Emissions: Current Status and Future Perspectives"

_ijerph, 2020, doi:10.3390/ijerph17197077_

Round 1

Reviewer 1 Report

This manuscript reviews academic papers on household CO2 emissions published between 1991 and 2020 to consider future research prospects. The authors collected bibliographic information of an enormous number of papers using a database search tool (Web-of-Science platform) and summarized the characteristics of past research activities. The conclusions derived from the review may not be new to climate mitigation professionals, but the collected data are beneficial to environmental researchers and policymakers. Please find the Reviewer's comments below.

Comment #1 (page 2, lines 54-63): As pointed out by the authors, the definition of household CO2 emissions is different from study to study, making it difficult to compare research results. A globally standardized rule (line 60) may mitigate this problem. However, creating such a rule would be difficult because a preferable definition of household CO2 emissions varies depending on research purposes and data availability. Do the authors have any idea about the standardization of household CO2 emissions? Please discuss it.

Comment #2 (page 2, lines 47-50): The authors show the proportion of household CO2 emissions to the national CO2 emissions for the USA, China, Canada, and Japan. Japan's emissions rate shown by the authors seems to be too high. In Japan, the Greenhouse Gas Inventory Office (GIO) calculates GHG emissions by sector and submits the emissions data to the UNFCCC every year (http://www.nies.go.jp/gio/en/aboutghg/index.html). According to the GIO data, the rate of household CO2 emissions (including indirect CO2 emissions from the use of electricity and heat) was 13.2%, 15.8%, and 14.6% in 2005, 2013, and 2018, respectively. These emissions rates are much lower than the authors' data (approximately 40%). Is the definition of household CO2 emissions proposed by the paper [13] different from the GIO?

Comment #3 (page 2, line 50): The authors cite the paper [14] to show Japan's emissions rate, but the paper seems not to include any information about Japan. Please check it.

Comment #4 (page 4, Figure 1): Figure 1 shows the time variation of the number of academic papers on household CO2 emissions. According to Figure 1, the number of papers published in the 1990s is quite small. Do the publication data collected from the WOS platform reflect actual research activities? Is it possible to search the papers published before online publication services became popular?

Comment #5 (page 6, lines 179-193): Figure 3 illustrates international cooperation relationships observed in the studies on household CO2 emissions. From this result, the authors concluded that cooperation with leading countries (the USA, UK, and Australia) is necessary to increase China's academic presence. According to Figure 3, however, most of the studies conducted in each country have no cooperation with other countries. Household CO2 emissions strongly depend on the characteristics of local communities (e.g., income level, climate conditions, and lifestyle), and therefore research results in a region cannot be directly applied to other regions. Did the authors find a good example of international cooperation through this review? Please discuss the benefits of international cooperation by citing some papers.

Comment #6 (page 7-8, lines 209-236): This manuscript uses bibliographic information on academic papers to consider future research prospects. The summary of bibliographic information provided by the authors is interesting but pays little attention to the dynamics of research activities. For example, the authors visualize the co-occurrence relationships between the keywords attached to the published papers (Figure 5) and discuss future research trends. In my opinion, however, Figure 5 indicates only the well-known fact that general keywords (e.g., energy, model, and efficiency) co-occur with high frequency. It is difficult to predict future research trends from this result. The Reviewer recommends that the authors investigate how the co-occurrence network structure changed with time.

Author Response

Point 1: As pointed out by the authors, the definition of household CO2 emissions is different from study to study, making it difficult to compare research results. A globally standardized rule (line 60) may mitigate this problem. However, creating such a rule would be difficult because a preferable definition of household CO2 emissions varies depending on research purposes and data availability. Do the authors have any idea about the standardization of household CO2 emissions? Please discuss it.

Response 1: Thank you for this thoughtful comment. We have discussed this point in the discussion part. The revision is shown below:

Second, a globally standardized rule is necessary for different expressions of HCEs. As mentioned above, the definitions of HCEs are different, especially based on the micro scale, which makes it difficult to compare research results. Creating such a rule is necessary for different research purposes and data availability, for example, by reviewing the HCEs literature of the international community and establishing the definition and boundary. As shown in Figure 6, HCEs are classified based on different energy types, different life demands and different consumption behaviors. Accordingly, a simple and practical HCEs calculator based on different levels for diverse stakeholders is needed.

Point 2: The authors show the proportion of household CO2 emissions to the national CO2 emissions for the USA, China, Canada, and Japan. Japan's emissions rate shown by the authors seems to be too high. In Japan, the Greenhouse Gas Inventory Office (GIO) calculates GHG emissions by sector and submits the emissions data to the UNFCCC every year (http://www.nies.go.jp/gio/en/aboutghg/index.html). According to the GIO data, the rate of household CO2 emissions (including indirect CO2 emissions from the use of electricity and heat) was 13.2%, 15.8%, and 14.6% in 2005, 2013, and 2018, respectively. These emissions rates are much lower than the authors' data (approximately 40%). Is the definition of household CO2 emissions proposed by the paper [13] different from the GIO?

Response 2: Thank you for this valuable and useful comment. We have changed the expression of this sentence. The revision is shown below:

In 2018, such emissions accounted for approximately 40% of total emissions in Japan [13, 14], but direct HCEs accounted for only 4.84% (there were 1078.03 million tons CO2 emissions in Japan in 2018, and 52.15 million tons CO2 emissions were from the residential sector) [15].

Point 3: The authors cite the paper [14] to show Japan's emissions rate, but the paper seems not to include any information about Japan. Please check it.

Response 3: Thank you for this careful attention. We marked the citing incorrectly before. We have changed this citing. Thanks again for the reviewer’s careful observation. We will be more careful within these details in the future.

Point 4: Figure 1 shows the time variation of the number of academic papers on household CO2 emissions. According to Figure 1, the number of papers published in the 1990s is quite small. Do the publication data collected from the WOS platform reflect actual research activities? Is it possible to search the papers published before online publication services became popular?

Response 4: Thank you for this valuable and useful comment. Web of Science (WOS) platform is an authoritative citation database source which contains thousands of international academic journals with multidisciplinary, comprehensive and high-impact. Data on HCEs publications are collected from the WOS platform, which is relatively popular and comprehensive. This work analyse the overall publishing trend, the contributions of categories, journals, countries/territories and institutions, as well as the topical subjects of HCEs to explore its progress and frontiers. However, this work has some limitations that can be improved upon in the future. For example, this work mainly includes literature from WOS platform in English, which resulted in an incomplete analysis. Future work should consider and compare non–English literature from different platforms, such as WOS, Scopus and others. We discuss these limitations in this work. We give a discussion which is shown below:

The present study uses bibliometric analysis on the progress of HCE research over the past thirty years. The HCEs publication data were collected from the WOS platform, which is relatively popular and comprehensive. This work analyzes the overall publishing trend, the contributions of categories, journals, countries/territories and institutions, and the topical subjects of HCEs research to explore the progress and frontiers of such research. However, this work has some limitations that can be improved upon in the future. For example, this work mainly includes literature from the WOS platform in English, resulting in an incomplete analysis. Future work should consider and compare non–English literature from different platforms, such as WOS, Scopus and others. Additionally, the factors of total citation frequency and citations per paper, which are not relevant to the topic of this paper, are not considered. How can a clearer theoretical framework be established in this field, and what is the general research paradigm for such a framework?

Point 5: Figure 3 illustrates international cooperation relationships observed in the studies on household CO2 emissions. From this result, the authors concluded that cooperation with leading countries (the USA, UK, and Australia) is necessary to increase China's academic presence. According to Figure 3, however, most of the studies conducted in each country have no cooperation with other countries. Household CO2 emissions strongly depend on the characteristics of local communities (e.g., income level, climate conditions, and lifestyle), and therefore research results in a region cannot be directly applied to other regions. Did the authors find a good example of international cooperation through this review? Please discuss the benefits of international cooperation by citing some papers.

Response 5: Thanks for reviewer’s attentions and comments. We give two examples of international cooperation benefit the academic research based on the references. The revision is shown as below:

It is necessary to consider cooperation between developing and developed countries in academic research, for example, on carbon labeling schemes [30], and market mechanisms [31]. Thus, there is an urgent need to strengthen cooperation between China and other countries, especially the USA, the UK, and Australia, to enhance China’s influence on HCEs research.

Point 6: This manuscript uses bibliographic information on academic papers to consider future research prospects. The summary of bibliographic information provided by the authors is interesting but pays little attention to the dynamics of research activities. For example, the authors visualize the co-occurrence relationships between the keywords attached to the published papers (Figure 5) and discuss future research trends. In my opinion, however, Figure 5 indicates only the well-known fact that general keywords (e.g., energy, model, and efficiency) co-occur with high frequency. It is difficult to predict future research trends from this result. The Reviewer recommends that the authors investigate how the co-occurrence network structure changed with time.

Response 6: Thank you for this valuable and useful comment. We give a figure about the keywords ranked by burst detection. The revision is shown as below:

Burst detection is used to determine whether there has been any change in the research hotspots, and it can help gain insights into future research topics [30, 32]. We give the keywords ranked by burst detection (the red rectangle means the strongest bursts), as shown in Figure 5(b). In this study, there are 25 keywords with apparent bursts. This result implies that HCEs research has been distinguished by three stages: in the first stage (2001-2010), as the infancy of HCEs research, the burst keywords mainly included energy use, CO2 emissions, industrial ecology, climate change, cost, household consumption, energy requirement, sustainable consumption, electricity, India, the USA and energy demand. In particular, HCEs research was gradually extended to direct energy usage and indirect household consumption to evaluate emissions. In the second stage (2012-2014), the main keywords were industry, trade, input-output analysis, renewable energy, simulation, technology, the UK, construction and environmental impact. In this stage, studies focused on HCEs methods and the possible impact of HCEs on the environment as well as the related influencing factors, such as trade, technology, and renewable energy usage. The third stage, from 2016 to 2018, involved research on HCEs at the city level and focused on sustainability and urbanization. HCEs research shifted from the whole country (such as in India, the USA, the UK) scale to the micro scale, such as the city level.

Reviewer 2 Report

The paper titled, “Household CO2 Emissions: Research Situation, Review and Prospect” reviews and projects household carbon emissions using bibliometric analysis and a systematic review based on the data available on Web of Science from 1991 to 2020. Overall, the paper seems fit for acceptance in the journal. The paper meets the scope and has a sound methodology along with suitable structure, language, and format according to the requirements of the Journal. However, before considering publishing the following comments should be addressed.

1: Citation is missing in many places. Provide the citation for page # 3, paragraph #2 "Published papers on the research of HCEs show an upward trend indicating ......

2. Page #3, paragraph # 3 "The papers published in 2020 were relatively new (the paper retrieval time was up to 107 July 2020),... provide the citation.

3. Page #5, paragraph 1 it is mentioned that the number of publications in the USA and China is significantly higher than that in other countries. But the author didn't provide the reason why the ratio of publication in these two countries is higher than in other countries.

4. Provide a table that gave a comparison of research on HCEs shifts from a macro level to a micro level between different territories from 1991 to 2020.

5. Improve the language of the paper.

Author Response

Point 1: Citation is missing in many places. Provide the citation for page # 3, paragraph #2 "Published papers on the research of HCEs show an upward trend indicating

Response 1: Thank you for this valuable and useful comment. We have provided the related citation. The revisions are shown below:

The number of published papers on HCEs shows an upward trend, indicating that awareness of HCEs has been growing [28, 29].

  1. Zhang, X.L.; Luo, L.Z.; Skitmore, M. Household carbon emission research: An analytical review of measurement, influencing factors and mitigation prospects. J. Clean. Prod. 2015, 103, 873–883.
  2. Geng, Y.; Chen, W.; Liu, Z.; et al. A bibliometric review: Energy consumption and greenhouse gas emissions in the residential sector. J. Clean. Prod. 2017, 159, 301–316.

Point 2: Page #3, paragraph # 3 "The papers published in 2020 were relatively new (the paper retrieval time was up to July 2020) provide the citation.

Response 2: Thank you for this valuable and useful comment. The research data were downloaded in July 2020, which caused a certain lag phenomenon. However, from the linear growth trend of the number of annual articles from 1991 to 2020, this phenomenon does not affect the overall analysis. Therefore, we changed the expression of the previous sentence, and the revisions are shown below:

However, this aspect does not affect the overall analysis because a linear growth trend (with R2=0.9104) shown based on the number of annual articles from 1991 to 2020.

Point 3: Page #5, paragraph 1 it is mentioned that the number of publications in the USA and China is significantly higher than that in other countries. But the author didn't provide the reason why the ratio of publication in these two countries is higher than in other countries.

Response 3: Thank you for this valuable and useful comment. We have given some reasons, and the revisions are shown below:

The number of publications by the USA (as a developed country) and China (as a developing country) is significantly higher than that by other countries. The main reason is that rapid economic development in the USA and China has led to improvements in quality of life as well as increased energy usage and the related HCEs. China published HCEs papers later, but it has rapidly developed and is now the country with the most publications. After 2012, the number of published articles in China surpassed that from the USA, with the significant advantage of being a latecomer. Additionally, HCEs accounted for >80% of the total emissions in the USA [9] and 30%–40% in China [10, 11], and CO2 emissions from the household sector have aroused widespread concern abroad [28]. Reducing HCEs will make a significant contribution to global climate action [29]. The USA, China, the UK and Australia are the main regions where HCEs research is active.

Point 4: Provide a table that gave a comparison of research on HCEs shifts from a macro level to a micro level between different territories from 1991 to 2020.

Response 4: Thank you for this valuable and useful comment. We provide a figure for the keywords ranked by burst detection. From the burst detection of the keywords, we find that from 2016 to 2018, research on HCEs is at the city level and focuses on sustainability and urbanization. We hope that this analysis may answer your question. The revision is shown below:

Burst detection is used to determine whether there has been any change in the research hotspots, and it can help gain insights into future research topics [30, 32]. We give the keywords ranked by burst detection (the red rectangle means the strongest bursts), as shown in Figure 5(b). In this study, there are 25 keywords with apparent bursts. This result implies that HCEs research has been distinguished by three stages: in the first stage (2001-2010), as the infancy of HCEs research, the burst keywords mainly included energy use, CO2 emissions, industrial ecology, climate change, cost, household consumption, energy requirement, sustainable consumption, electricity, India, the USA and energy demand. In particular, HCEs research was gradually extended to direct energy usage and indirect household consumption to evaluate emissions. In the second stage (2012-2014), the main keywords were industry, trade, input-output analysis, renewable energy, simulation, technology, the UK, construction and environmental impact. In this stage, studies focused on HCEs methods and the possible impact of HCEs on the environment as well as the related influencing factors, such as trade, technology, and renewable energy usage. The third stage, from 2016 to 2018, involved research on HCEs at the city level and focused on sustainability and urbanization. HCEs research shifted from the whole country (such as in India, the USA, the UK) scale to the micro scale, such as the city level.

Additionally, research on HCEs at the microlevel scale needs to pay attention to the following three aspects. First, the microlevel calculation model for HCEs needs to be optimized and modified, even though the mainstream methods of analyzing HCEs are very mature. This need mainly stems from the large differences in the technological level and production capacity of different countries and regions, resulting in inconsistent emission factors in the production and consumption of fossil fuels [66]. When calculating HCEs on a micro scale, emission factors need to be considered to make the calculated data more reasonable and comparable [67]. Second, a globally standardized rule is necessary for different expressions of HCEs. As mentioned above, the definitions of HCEs are different, especially based on the micro scale, which makes it difficult to compare research results. Creating such a rule is necessary for different research purposes and data availability, for example, by reviewing the HCEs literature of the international community and establishing the definition and boundary. As shown in Figure 6, HCEs are classified based on different energy types, different life demands and different consumption behaviors. Accordingly, a simple and practical HCEs calculator based on different levels for diverse stakeholders is needed. Third, the research perspectives on HCEs need to be refined, and data need to be mined. Against the backdrop of big data, a wide variety of data, such as remote sensing satellite data (Landsat data, data from the Defense Meteorological Satellite Program's Operational Linescan System (DMSP/OLS), data from the Suomi National Polar-Orbiting Partnership’s Visible-Infrared Imaging Radiometer Suite (NPP/VIIRS), etc.) [68, 69], statistical data (statistical yearbooks, statistics, etc.) [10, 70], and survey data [20, 22] are used to research HCEs. As a result, HCEs data are diverse, complicated, and heterogeneous and requires in–depth analysis and data mining.

Point 5: Improve the language of the paper.

Response 5: English-language diction, syntax, and grammar have been edited by “SPRING NATURE Author Services (https://authorservices.springernature.com/language-editing/)”. Some grammatical mistakes in this study have already been corrected. We do not show here the one-by-one revisions that were made, but they are clearly highlighted in red.

Round 2

Reviewer 1 Report

The authors properly responded to the Reviewer’s comments, and the manuscript was much improved. The Reviewer concluded that the revised version of the manuscript is suitable for publication in International Journal of Environmental Research and Public Health.